# Using moral foundations in government communication to reduce vaccine hesitancy

**Florian Heine** [ID] *, Ennie Wolters

Tilburg University, Tilburg School of Economics and Management, Tilburg, The Netherlands

* f.a.heine@tilburguniversity.edu

## Abstract

Having a vaccine available does not necessarily imply that it will be used. Indeed, uptake rates for existing vaccines against infectious diseases have been fluctuating in recent years. Literature suggests that vaccine hesitancy may be grounded in deeply rooted intuitions or values, which can be modelled using Moral Foundations Theory (MFT). We examine the respective prominence of the MFT dimensions in government communication regarding childhood vaccinations and explore its effect on parents' vaccine hesitancy. We measure the MFT dimension loading of the vaccination information brochures from the Dutch National Institute for Public Health and the Environment (RIVM) between 2011-2019 and connect this information with the electronic national immunisation register to investigate if the use of moral foundations in government communication has a measurable effect on vaccination uptake. We find the largest positive effect for the dimensions Authority/Subversion and Liberty/Oppression and suggestive evidence in favour of a small positive effect for Purity/Degradation. Conversely, Loyalty/Betrayal actually has a negative effect on vaccination rates. For the dimension Harm/Care, we find no significant effect. While Purity/Degradation and Harm/Care appear to be the two most frequently used moral foundations by RIVM, these dimensions have in fact no or only a minor effect on parents' vaccine hesitancy. Reducing the use of these moral foundations may be the first step towards optimising government communication in this context. Instead, formulations activating the moral foundations Authority/Subversion and Liberty/Oppression appear to have positive effects on vaccination uptake.

## Introduction

One of the main societal motivations for childhood vaccinations is to extinguish infectious diseases, such as Measles, Tetanus and Polio, by achieving herd immunisation [1]. Having a vaccination available, however, does not automatically mean that it will be used [2–5]. Advancing the uptake of the vaccine among the public is one of the main challenges for policymakers [6]. In the context of the current COVID-19 pandemic with its widespread vaccination programmes, the issue of vaccine hesitancy has gained new relevance [7]. Understanding factors influencing vaccination acceptance rates can be of crucial importance for the success of this

**Data Availability Statement:** All data files are available from the DANS database (DOI: 10.17026/dans-xmn-mtdn).

**Funding:** The author(s) received no specific funding for this work.

**Competing interests:** The authors have declared that no competing interests exist.

and other vaccination programmes [8, 9]. Thomson and Watson [10] suggest vaccination adoption to be an additively separable function of *access* and *acceptance* and coin the axiom "vaccine adoption = access+acceptance" [10]. By using data from the Netherlands, a country in which childhood vaccinations are free of charge and readily accessible for everyone, we hold the aspect of *access* constant to investigate *acceptance*-driven parental choice in vaccine adoption rates. Prior studies have mainly focused on the aspect of parental beliefs about childhood vaccinations to explain vaccination uptake *intentions* (i.e. [11–15]). These are often collected as self-stated vaccination intention through surveys, which may not necessarily translate into actual vaccination uptake (see, e.g. [16]). In this study, by contrast, we employ vaccine adoption rates from governmental care records, which has been characterised as the "gold standard" [17] to measure actual behaviour as opposed to self-stated vaccine hesitancy. Prior evidence-based studies on vaccine hesitancy have focused predominantly on raising knowledge and awareness [18]. Most interventions demonstrate only short-term effects or even a decrease in intention to vaccinate, though [19, 20]. This resistance towards educational interventions suggests that attitudes towards vaccines may instead be rooted in deeper intuitions and emotions.

Moral Foundations Theory was developed to identify "intuitive ethics" that guide people's behaviour [21]. It consists of six dimensions: Care/Harm, Fairness/Cheating, Loyalty/Betrayal, Authority/Subversion, Purity/Degradation and Liberty/Oppression [22]. We discuss the concept of Moral Foundations Theory in more detail in S1 Appendix. MFT is a relatively new theory, developed at first to explain the dimensions of moral judgement, which has been shown to be an intuitive process [22]. Since then, moral foundations have been applied to explain general political ideologies [23], or specific political stances, for example on death penalty, abortion, gun control and immigration [24], but also on attitudes towards climate change [25], and leadership moralisation [26]. Clifford and Jerit [27] show how the use of moral rhetoric by political elites in the debate about stem cell research affects public attitudes. In this study we investigate if government communication has a measurable effect on parents' choice to vaccinate their child and most importantly, which moral dimensions appear to trigger these effects.

Amin et al. [28] have been the first to apply MFT in the context of vaccination uptake. They conduct a survey using MFT to explain how deeply rooted intuitions, or "values", influence vaccine hesitancy. They find that "medium vaccine hesitant parents" are twice as likely to emphasise the moral foundation Purity/Degradation and "very vaccine hesitant parents" are twice as likely to emphasise Purity/Degradation and Liberty/Oppression. The authors suggest that government communication about childhood vaccinations might be more effective in reaching these vaccine hesitant parents if it focuses on Purity/Degradation and Liberty/Oppression. To date, there exists no evidence on whether triggering specific moral foundations translates into measurable change in behaviour, though. Our study contributes at filling this knowledge gap by linking the MFT dimensions used in vaccine related government communication towards parents with actual governmental care records on vaccination uptake.

We use the Moral Foundations Dictionary (MFD) [23, 29, 30], a validated and established dictionary providing information on the words for each foundation, to analyse all vaccination information brochures directed at parents by the Dutch National Institute for Public Health and the Environment (RIVM) between 2011 and 2019. In the Netherlands, care providers hand out these brochures during one of various mandatory consultation visits to *all* parents with children in the appropriate age group for a given vaccination. We find that RIVM has mainly used the moral foundations Purity/Degradation and Harm/Care in their communication towards parents, while the dimensions Authority/Subversion and Fairness/Cheating have found the least use. In fact, Fairness/Cheating has not been used at all in RIVM's brochures. We then connect this information on MFT use in the vaccination brochures with vaccination

rates from the electronic national immunisation register (Præventis). We find robust evidence for a favourable effect in the moral foundations Authority/Subversion and Liberty/Oppression for reducing parents' vaccine hesitancy. By contrast, the use of the moral foundation Loyalty/Betrayal in the brochures appears to translate into a negative effect on vaccination rates.

This article is structured as follows: *First* we introduce our methods. We briefly outline the Dutch Immunisation Programme (NIP), including details on the use of vaccination brochures, before describing our data collection method. *Then* in Section Results we discuss our results. More concretely, we analyse the use of MFT dimensions in government issued vaccination brochures and then use this information to examine the relationship between the MFT dimensions and vaccination uptake rates. *Finally*, we will discuss the implications of our results and the associated limitations of our research design in Section Discussion and Conclusion.

## Methods

We study childhood vaccination adoption in the Netherlands as a function of moral rhetoric in government communication. First, in Subsection The Dutch National Immunisation Programme, we give a very brief overview of the Dutch National Immunisation Programme (NIP). The first step in our analysis is to code all RIVM brochures on childhood vaccinations targeted at parents between 2011 and 2019. We describe this step in Subsection RIVM Childhood Vaccination Brochures and Moral Foundations. Finally, we use this information—and other control factors—as independent variables to explain the vaccination uptake of various vaccinations within the Dutch NIP during the period 2011–2019. We provide a brief discussion of the national immunisation register and the underlying data in Subsection Immunisation Coverage and Vaccine Hesitancy.

### The Dutch National Immunisation Programme

The Dutch National Immunisation Programme (NIP) started in 1957 by offering vaccinations for Diphtheria, Tetanus and Polio [31–35]; more vaccinations have been added since. We include the vaccination schedule of the Dutch NIP applicable for the period 2011–2019 in Fig 1, including an indication of the age at which a child should receive certain vaccinations. The Figure also includes a legend with abbreviations which we use in this article.

Vaccinations have been a 'hot topic' in Dutch society in recent years [35]. There have been debates about vaccine safety, the non-mandatory status of vaccinations, the admission of non-vaccinated children to kindergarten and a need for improvement of government communication to increase the immunisation coverage [37]. While the immunisation coverage in the Netherlands has been high, historically, RIVM has documented a 2–3% decrease since 2014 [35]. In the Netherlands, immunisation coverage has reached a level below 95%, a percentage that the WHO uses as a threshold for the successful achievement of herd immunisation [38].

### RIVM childhood vaccination brochures and moral foundations

To measure the use of moral foundations in government communication, we analyse brochures on childhood vaccinations which are provided by the National Institute for Public Health and the Environment (RIVM) to all parents. These brochures constitute the main communication channel for RIVM to provide parents with information about the Dutch NIP. Children ought to receive one or two vaccinations at a time during various points in their youth (see Fig 1). All vaccinations mentioned in the Dutch NIP are on a voluntary basis and free of charge [35]. Hence, it is the parents who freely decide whether they want to vaccinate their children. However, the government, represented by RIVM, does promote vaccinating children by providing parents with information brochures about the upcoming vaccination

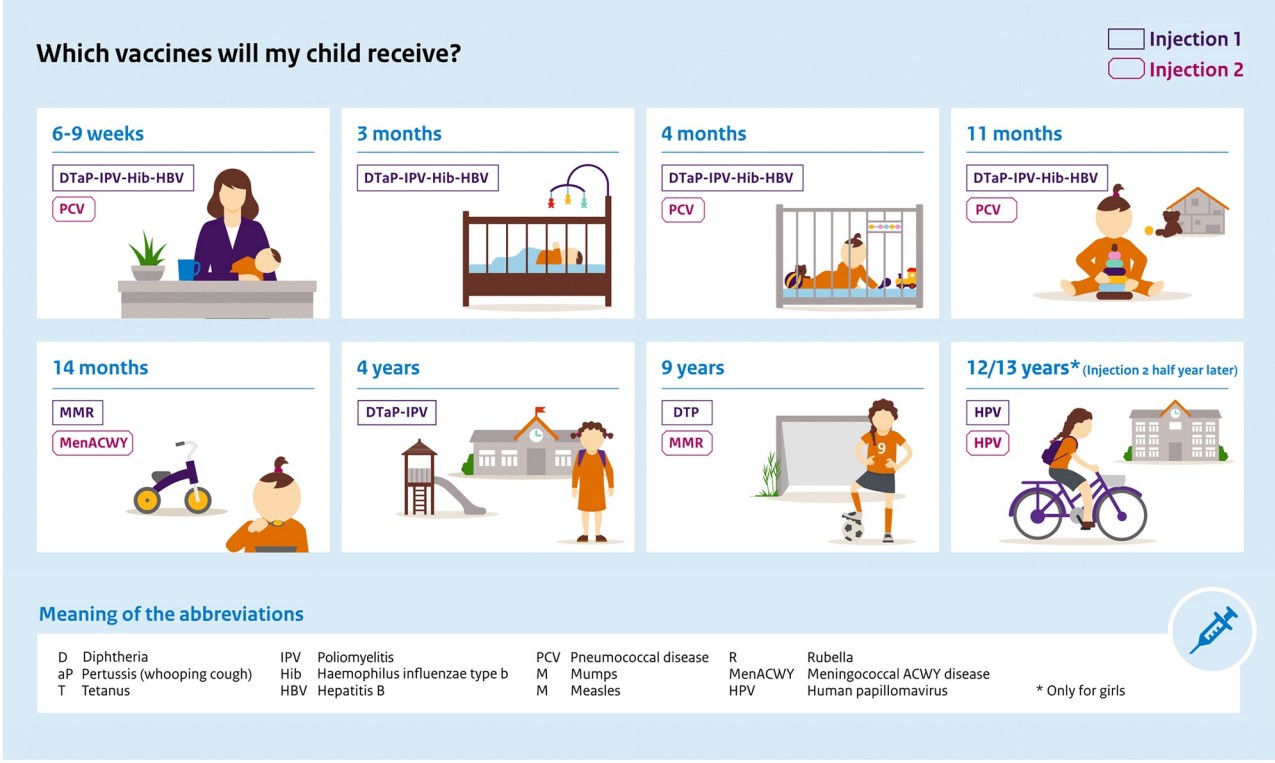

**Fig 1. Vaccination schedule 2011–2019.** Source: Rijksinstituut voor Volksgezondheid en Milieu [36], with permission.

and the Dutch NIP. In chronological order, this process of information provision and parents' choice proceeds as follows:

1. RIVM writes brochures containing information about childhood vaccinations mentioned in the Dutch NIP.

2. A child approaches a certain age at which, according to the Dutch NIP, he or she should be vaccinated (i.e.: at birth, before 3 months of age, before 4 years of age. . .).

3. Parents go to see the midwife (before giving birth), the consultation clinic (small children) or the Municipal Public Health Services, GGD (9 years old and adolescent girls). These visits are *mandatory* in the Netherlands.

4. During these visits, parents receive the brochures about the vaccinations mentioned in the Dutch NIP that correspond to the child's age. A parent receives a total of four to five brochures (depending on the child's sex) over the time span of parenting a child.

5. Parents go home, read the brochure and choose whether or not they want to vaccinate their child.

6. Parents receive an invitation letter/call to vaccinate their child at a location near them.

7. Parents do or do not vaccinate their child.

RIVM uses different brochures for parents of children of different age groups, and these brochures refer to the specific vaccinations that are linked to that age category. In other words, a given brochure refers to the vaccinations which the child, according to the vaccination schedule of the Dutch NIP, should receive shortly. To illustrate: the brochure for parents of toddlers refers to the vaccination DTaP-IPV and the brochure for schoolchildren refers to the DTP-IPV and MMR vaccinations. In Table 1, we present an overview of the vaccinations mentioned in the Dutch NIP and the corresponding brochures that contain information about these vaccinations (original Dutch names).

As RIVM does not publish a new brochure every year, we always use the latest edition of a brochure that RIVM has published, which corresponds to the actual use by RIVM. To illustrate: if RIVM published a new brochure in 2012 but no new brochure in 2013, it uses the 2012 brochure for 2013. We present an overview of these brochures and the associated vaccinations in Table 2.

Note that for most vaccinations, our data covers the years 2011–2019, with four exceptions. *1*) The data for the DT-IPV schoolchildren vaccination only covers the years 2012–2019, because RIVM first created a brochure for this vaccination in 2012. *2*) The same also holds for MMR schoolchildren. This brochure has been first published and distributed together with the one for DT-IPV schoolchildren, hence also for this vaccination, the data covers the years 2012–2019. *3*) The data for the HPV vaccination for adolescent girls only covers the years 2013–2019, because the relevant brochure was only introduced in 2013 [31]. *4*) Finally, the data for the HBVa vaccination only covers 2015–2019 because of a new regulation and implementation issues abound. Before 2011, only risk groups were vaccinated against HBVa. From 2011 onward, however, the HBVa vaccination was implemented for the entire cohort [31]. The effects of this new regulation only manifest from 2014 onward, though, as 2011 is the cohort for the immunisation coverage of HBVa in 2014. We exclude 2014, the first year, because the immunisation coverage in 2014 was only 51.4% before 'stabilising' between 2015 and 2019 around 90+% (see S1 Table). It is highly unlikely that the increase in immunisation coverage from 2014 to 2015 is caused by the use of moral foundations in brochures. It is more likely that the low immunisation coverage in 2014 has to do with start-up issues.

**Table 1. Vaccinations and corresponding brochures.**

| Vaccination | Corresponding brochure (original Dutch title) |
|---|---|
| **Newborns** | |
| DTaP-IPV newborns | Folder babies |
| Hib | Folder babies |
| HBVa | Folder babies |
| PCV | Folder babies |
| MenC | Folder peuters |
| MMR newborns | Folder peuters |
| **Toddlers** | |
| DTaP-IPV toddlers | Folder kinderen 4 jaar |
| **Schoolchildren** | |
| DT-IPV schoolchildren | Folder kinderen 9 jaar |
| MMR schoolchildren | Folder kinderen 9 jaar |
| **Adolescent girls** | |
| HPV | Folder HPV |

**Table 2. Overview of the brochures per year and the vaccinations mentioned in each brochure.**

| Brochures per year (original Dutch name) | Vaccinations mentioned in brochure |
| --- | --- |
| **2011** | |
| Folder baby's van 2, 3, 4 en 11 maanden 2011 | DTaP-IPV newborns, Hib, HBVa, PCV |
| Folder kinderen 4 jaar 2011 | DTaP-IPV toddlers |
| Folder peuters 14 maanden 2011 | MenC, MMR newborns |
| **2012** | |
| Folder baby's 2, 3, 4 en 11 maanden 2012 | DTaP-IPV newborns, Hib, HBVa, PCV |
| Folder kinderen 4 jaar 2012 | DTaP-IPV toddlers |
| Folder kinderen 9 jaar 2012 | DT-IPV, MMR schoolchildren |
| Folder peuters 14 maanden 2012 | MenC, MMR newborns |
| **2013** | |
| Folder baby's van 6–9 weken, 3, 4 en 11 maanden 2013 | DTaP-IPV newborns, Hib, HBVa, PCV |
| Folder extra BMR baby's | MMR |
| van 6–14 maanden 2013 | |
| Folder HPV 2013 | HPV |
| Folder kinderen 9 jaar 2013 | DT-IPV, MMR schoolchildren |
| **2014** | |
| Folder HPV 2014 | HPV |
| **2015** | |
| Folder baby's 6–9 weken 2015 | DTaP-IPV newborns, Hib, HBVa, PCV |
| Folder HPV 2015 | HPV |
| Folder kinderen 4 jaar 2015 | DTaP-IPV toddlers |
| Folder kinderen 9 jaar 2015 | DT-IPV, MMR schoolchildren |
| Folder peuters 14 maanden 2015 | MenC, MMR newborns |
| **2016** | |
| Folder HPV 2016 | HPV |
| **2018** | |
| Folder HPV 2018 | HPV |
| Vaccinaties voor kinderen van 4 jaar 2018 | DTaP-IPV toddlers |
| Vaccinaties voor kinderen van 9 jaar 2018 | DT-IPV, MMR schoolchildren |
| **2019** | |
| Vaccinaties voor kinderen van 9 jaar 2019 | DT-IPV, MMR schoolchildren |

We then use the Moral Foundations Dictionary (MFD) [23, 29], which is a list of words and word stems developed for word analysis purposes, to measure the use of moral foundations in these RIVM brochures. For each moral foundation, the MFD includes a list of words and word stems relating to that specific moral foundation (We will include an overview of the MFD in an online repository). As the original MFD does not contain the sixth moral foundation 'Liberty/Oppression', we use the list of indicating words for Liberty/Oppression created by Teernstra et al. [30], who analyse tweets on the five original moral foundations and the sixth foundation Liberty/Oppression. Their list is the most established tool in the field for analysing texts on the moral foundation Liberty/Oppression (We include this list in the online S1 Appendix). The MFD and the list of indicating words for Liberty/Oppression are originally written in English. As the brochures from RIVM are written in Dutch, we translate the English Moral Foundations Dictionary and the list of indicating words for the foundation Liberty/Oppression to Dutch. Similar to the method outlined by Harzing [39], both authors who are fluent/native Dutch speakers, independently translate the original English MFD words and word stems into Dutch and resolve any differences in the separately generated translations by

discussion. We include this 'Dutch MFD' in Supporting Information S1 File and example sentences for each MFT dimension in Table A3 in S1 Appendix. While we remain within the same language family branch (Germanic) for this translation from English to Dutch, residual differences in meaning could potentially account for part of our obtained results. This is a potential limitation which our study shares with a large class of research in a cross cultural context that relies on translating instruments [40, 41].

To measure the use of moral foundations, we scan the brochures from RIVM for words included in the Dutch MFD using the software Linguistic Inquiry Word Count Program (LIWC), which was also used by Clifford and Jerit [27] and Teernstra et al. [30]. The software analyses texts for words indicated by the user of the programme and produces a measure of the extent to which each of the moral foundations is used in the brochure. Specifically, LIWC produces variables expressed in word counts. These variables refer to percentages of words relating to a specific moral foundation in a document, in this case a brochure. For example, if for a hypothetical Brochure *A*, LIWC indicates a value of 1.58 for Liberty/Oppression, this means that 1.58% of the words in Brochure *A* refer to the moral foundation Liberty/Oppression. Fig 2 provides a visual overview of the LIWC word counts per MFT dimension (An overview of the output of LIWC including the use of moral foundations in the different brochures is included in the S1 Appendix).

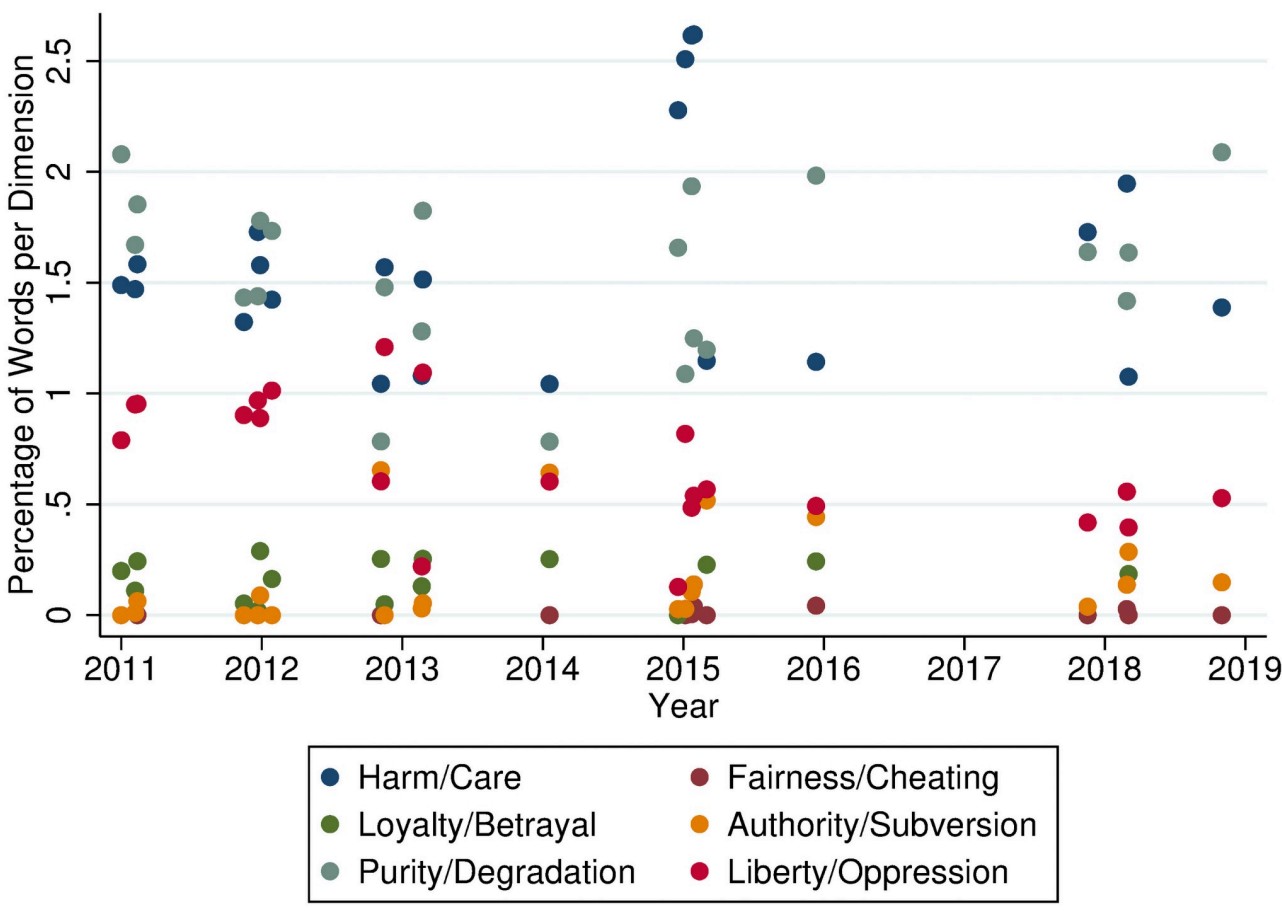

**Fig 2. Overview of the dimension loading of the RIVM brochures per year, jittered with 5% noise.**

### Immunisation coverage and vaccine hesitancy

We use the immunisation coverage (the proportion of the cohort vaccinated in percentages) of children to measure parents' vaccination acceptance. Immunisation coverage is inversely related to vaccine hesitancy, such that a higher immunisation coverage among children indicates lower vaccine hesitancy among parents. As opposed to self-stated vaccine hesitancy, the immunisation coverage represents an actual change in behaviour towards vaccinations, revealing parents' true vaccination acceptance. This change in behaviour is the essential goal of government communication: reducing vaccine hesitancy such that it actually leads to a higher immunisation coverage for childhood vaccinations.

We employ publicly available data on the immunisation coverage for the different vaccinations of interest provided by Statistics Netherlands (CBS) [42]. An overview of the *national* immunisation coverage per vaccination is included in S1 Table. The data provides information about the immunisation coverage of all vaccinations mentioned in the Dutch NIP over the years. The data originates from 'Præventis', which is an electronic national immunisation register, developed in 2005 [43]. Præventis is linked to the Dutch population register. It records and validates administered vaccinations at the individual level. Because of its link to the Dutch population register, it is safe to characterise the data to be very close or even equal to the actual population. We use this data at the national aggregate level, as well as at the level of the 25 municipal public health regions (GGD-regio's, see [44]) and the 355 municipalities [45] in the Netherlands.

## Results

We first present results on the use of moral foundations dimensions in Subsection The Use of Moral Foundations. We analyse if certain moral dimensions are used more frequently than others in the brochures of RIVM. Then, in Subsection The Effect of the Use of Moral Foundations on Vaccine Hesitancy, we use the dimension loading of the RIVM brochures to investigate if the use of moral foundations has a measurable effect on vaccination uptake.

### The use of moral foundations

We use the MFT dimension loading produced by the LIWC-coded word count to understand which moral foundations RIVM uses in its communication towards parents. Fig 2 provides a graphical overview of the dimension loading of each brochure in a jittered scatter plot illustrating the LIWC word count (i.e. the percentage of words per MFT dimension) as a function of the brochure's year of use. Each colour represents a different MFT dimension, so each published brochure is represented in the graph by six dots of different colour, representing one word count indication per dimension per brochure. Consider, for example, 2016. For this year we only have one new brochure, 'Folder HPV 2016'. Accordingly, Fig 2 depicts six different coloured data points for 2016. This initial eyeballing of the data gives a first impression of the heterogeneity of MFT dimensions used in RIVM's brochures. It appears that Harm/Care and Purity/Degradation are used more frequently than other dimensions. Equally, Fairness/Cheating appears to consistently score very low for the word count. In fact, this dimension has never been used in the brochures. In the following we quantify this heterogeneity. As the data is not normally distributed we employ the rank-based non-parametric Kruskal-Wallis test [46] to identify structural patterns in the use of MFT.

This test can indicate whether there is a significant difference in the use of moral foundation dimensions in RIVM's brochures. As it does not indicate the directionality of an eventual difference between the categories, we run a post-hoc Dunn's test (pairwise comparisons) with Benjamini-Hochberg correction for multiple hypotheses testing to see which moral foundations differ significantly from each other [47]. This test compares each category (a category is a

**Table 3. Output for the Kruskal-Wallis test.**

| Moral Foundations | N | Rank Sum |
|---|---|---|
| Purity/Degradation | 22 | 2428.5 |
| Harm/Care | 22 | 2403.5 |
| Liberty/Oppression | 22 | 1681.5 |
| Loyalty/Betrayal | 22 | 1061.0 |
| Authority/Subversion | 22 | 840.5 |
| Fairness/Cheating | 22 | 363.0 |

moral foundation dimension) to all other categories. The Dunn's test produces $Z$-test statistics for each pairwise comparison and the $p$-value [47].

The Kruskal-Wallis test indicates a significant difference in the use of moral foundations in RIVM's brochures across the six categories of moral foundations with $\chi^{2(5)} = 112.583$, Â $p = 0.0001$ for $\alpha = 0.05$. Table 3 provides a list of the rank sum for each of the MFT dimensions, ordered by most to least prominent.

Note that the MFT dimension Fairness/Cheating has in fact never been used in any of the brochures. The value for each of the observations was 0. This means that RIVM has never used words or sentences that relate to fairness, justice and trustworthiness. An example of such a sentence would be: *"This brochure contains fair and unprejudiced information about vaccinations."* Kennedy et al. [48] investigate the relationship between language usage and the five MFT dimensions by analysing Facebook status updates. Also in their study, Fairness/Cheating was the most difficult dimension to trace in language.

We use a post-hoc Dunn's test with Benjamini-Hochberg correction to describe which of the pairwise comparisons between the MFT dimension loadings can be described as significantly different from each other [47]. The results of the Dunn's test are displayed in Table 4. Most results were significant (for $\alpha = 0.05$). The results which were *not* significant are underlined.

The results of the post-hoc Dunn's test show that the use of moral foundations in RIVM's brochures is significantly different for almost all moral foundations dimensions at a $p$-value of $p < 0.05$. The only two individual categories that do *not* differ significantly are:

- Harm/Care compared to Purity/Degradation

- Loyalty/Betrayal compared to Authority/Subversion

**Table 4. Dunn's pairwise comparison of use of moral foundations by moral foundations (Benjamini-Hochberg).** Underlined results are *not* significant at 5% confidence level.

| Column mean—row mean z test statistic (p-value) | Harm/Care | Fairness/Cheating | Loyalty/Betrayal | Authority/Subversion | Purity/Degradation |
|---|---|---|---|---|---|
| **Fairness/Cheating** | 8.102478 | | | | |
| | 0.0000 | | | | |
| **Loyalty/Betrayal** | 5.330839 | -2.771639 | | | |
| | 0.0000 | 0.0038 | | | |
| **Authority/Subversion** | 6.206407 | -1.896071 | 0.875568 | | |
| | 0.0000 | 0.0334 | 0.2042 | | |
| **Purity/Degradation** | -0.099271 | -8.201749 | -5.430110 | -6.305678 | |
| | 0.4605 | 0.0000 | 0.0000 | 0.0000 | |
| **Liberty/Oppression** | 2.866939 | -5.235539 | -2.463900 | -3.339468 | 2.966210 |
| | 0.0031 | 0.0000 | 0.0086 | 0.0008 | 0.0025 |

## The effect of the use of moral foundations on vaccine hesitancy

Using OLS with time-demeaning fixed effects, we investigate the relationship between parents' vaccine hesitancy, expressed by the immunisation coverage (data from Præventis and [42]), and the use of moral foundations in government communication (the output of LIWC's analysis of the brochures from RIVM). For this, we aim at explaining the immunisation coverage of the vaccinations mentioned in the Dutch National Immunisation Programme (NIP) (dependent variable / $Y$) by the use of moral foundations in government communication (independent variables / $X$'s) and controls.

As parents receive a given brochure only once (when their child is of a specific age), the use of moral foundations in the brochures that were used in the past does not affect the vaccine hesitancy of parents that, according to the Dutch NIP, should vaccinate their children the next year. In other words, the treated population is made up of different individuals each year, maintaining high temporal independence for our analysis.

Our data set contains the immunisation coverage ($Y_{ijt}$) for each vaccination $i$ in region $j$ and each year $t$, as well as the output from LIWC listing the use of moral foundations in the brochures from RIVM ($X$'s). As we have no observations for Fairness/Cheating, this dimension will not be part of the ensuing analysis. The data set can be accessed through the S1 File and example sentences for each MFT dimension in Table A3 in S1 Appendix and it is publicly available via [42]. Vaccination $i$ can be any of the type as in Table 1; region $j$ can either be referring to data on the national level (in this case we have $j = 1$ only), regional (with $j \in \{1, \ldots, 25\}$), or municipal level ($j \in \{1, \ldots, 355\}$). As discussed above, RIVM does not create a new brochure for each vaccination every year. For the years in which RIVM did not create a new brochure, we use the last brochure they published. This results in a complete data set, in which for each vaccination there is data on the use of moral foundations for every year. Next to vaccination rates for each vaccination type on a national aggregate, Præventis also provides this information for each of the 25 municipal public health regions (GGD-regio's, see [44]) and 355 municipalities [45] in the Netherlands. We complement an overall analysis using data aggregated at the national level, with more refined data at the regional and municipal level to demonstrate the sensitivity of our results. We control for population size (more accurately: The number of children in a given birth cohort, which should receive the specific vaccination in question, according to the Dutch NIP), which varies both by panel (i.e. region and vaccination type) and over time. Furthermore, we include time as control variable (2011 = 1...2019 = 9). The general model looks as follows:

$$
\begin{aligned}
\text{Immunisation Coverage}_{ijt} \;=\; & \beta_0 \\
+\; & \beta_1 \, HC_{it} \\
+\; & \beta_2 \, PD_{it} \\
+\; & \beta_3 \, LO_{it} \\
+\; & \beta_4 \, LB_{it} \\
+\; & \beta_5 \, AS_{it} \\
+\; & \beta_6 \, \text{Time}_t \\
+\; & \beta_7 \, \text{Population}_{ijt} + \varepsilon
\end{aligned}
\tag{1}
$$

where

| | |
|---|---|
| *HC* | is the score on the Harm/Care dimension |
| *PD* | is the score on the Purity/Degradation dimension |
| *LO* | is the score on the Liberty/Oppression dimension |
| *LB* | is the score on the Loyalty/Betrayal dimension |
| *AS* | is the score on the Authority/Subversion dimension |

To assess the robustness of our results, we add four additional controls to the model above. *1*) As proxy for the degree of difficulty of the language used, we include a count of words with six or more letters. *2*) As theoretically irrelevant structural composition element, we include the share of punctuation. *3*) To control for words referring to issues of general morality, we include a category from the MFD [29] which refers to generally morally loaded words that do not pertain to a specific MFT dimension. *4*) We account for the different length of brochures by including the absolute word count for each brochure.

Table 5 presents the results of the fixed-effects regressions, commensurable to Model 1. Regressions (1) and (2) use annual vaccination rates for the vaccinations of the Dutch NIP (as outlined in Subsection The Dutch National Immunisation Programme), aggregated at the national level; Regressions (3) and (4) are at the level of the 25 municipal health regions; and Regressions (5) and (6) use data at the level of all Dutch municipalities. While the uneven regressions represent results for the main model (Model 1), the even regressions include additional controls, as discussed above to assess the robustness of our results. Each combination of vaccination *i* and region *j* constitutes a separate panel. Within the Netherlands there exists some degree of regional variation as to the general vaccination acceptance, and each type of vaccination also displays some heterogeneity in acceptance rate. By demeaning these fixed effects, we are able to better isolate the true betas without the region- or vaccination specific fixed effects.

For three of the six moral foundations we find a robust significant effect on the immunisation coverage. Two of these have a positive effect on the immunisation coverage. These are respectively *Authority/Subversion* and *Liberty/Oppression*. *Loyalty/Betrayal*, by contrast has a negative effect on the immunisation coverage. The moral foundation *Purity/Degradation* displays a small positive effect which is significant in Regression (6) only. For this dimension, no significant effect emerges in the absence of our additional controls and for data at the national or regional level. The coefficient for *Harm/Care* is fairly volatile between the regressions and varies between about -2 and 0.7 percentage points. As the moral foundation *Fairness/Cheating* was never used in the brochures, it is discarded from the analysis.

Our results indicate a small but robust negative time trend ($\beta \leq -0.46, p < 0.001$), confirming the observation about the Dutch NIP discussed in Subsection The Dutch National Immunisation Programme of a slowly decreasing immunisation coverage over recent years [35]. Our regression output suggests a decrease of overall immunisation rates at a magnitude of about 0.46–0.74 percentage points annually.

**Main result 1**. The use of moral foundations in government communication affects vaccine hesitancy.

Overall speaking, the use of moral foundations in the RIVM's brochures does seem to affect the immunisation coverage ($p < 0.0001$). As displayed in Table 5, the models account for 26.6–73.5% of the variance within the independent variables. Amin et al. [28] find that vaccine hesitant individuals emphasise the moral foundations *Purity/Degradation* and *Liberty/*

**Table 5. Results of OLS fixed-effects models regressing the vaccination rate for vaccinations of the Dutch NIP on the MFT dimension loading from the associated information brochures and controls.** We analyse this for data reported at the national, regional and municipal level.

| | (1) | (2) | (3) | (4) | (5) | (6) |
|---|---|---|---|---|---|---|
| | National | | Regional | | Municipal | |
| **VARIABLES** | **Vaccination Rate** | | | | | |
| Purity/Degradation | 0.013 | 0.781 | -0.055 | 0.553 | 0.144 | 0.809*** |
| | (0.95) | (1.23) | (0.47) | (0.36) | (0.15) | (0.13) |
| Harm/Care | 0.687 | -1.789** | 0.425** | -2.048*** | 0.301*** | -2.008*** |
| | (0.65) | (0.77) | (0.19) | (0.22) | (0.08) | (0.11) |
| Liberty/Oppression | 4.098** | 3.890** | 4.303*** | 3.657*** | 4.296*** | 3.641*** |
| | (1.69) | (1.77) | (0.64) | (0.70) | (0.22) | (0.23) |
| Loyalty/Betrayal | -6.993 | -14.513*** | -6.142*** | -14.345*** | -5.848*** | -13.836*** |
| | (4.40) | (3.98) | (1.30) | (1.28) | (0.46) | (0.52) |
| Authority/Subversion | 22.452*** | 42.691*** | 22.568*** | 44.422*** | 23.047*** | 43.730*** |
| | (4.28) | (7.22) | (2.56) | (3.49) | (0.97) | (1.54) |
| Time | -0.460*** | -0.770*** | -0.564*** | -0.824*** | -0.561*** | -0.782*** |
| | (0.14) | (0.14) | (0.04) | (0.04) | (0.01) | (0.02) |
| Population | 0.000 | 0.000 | 0.001* | 0.000 | 0.002*** | 0.001*** |
| | (0.00) | (0.00) | (0.00) | (0.00) | (0.00) | (0.00) |
| Constant | 67.829*** | 67.175*** | 82.718*** | 75.669*** | 86.999*** | 76.333*** |
| | (14.08) | (18.87) | (3.40) | (3.42) | (0.53) | (1.11) |
| Controls | No | Yes | No | Yes | No | Yes |
| Number of observations | 82 | 82 | 2050 | 2050 | 28946 | 28946 |
| Number of panels | 10 | 10 | 250 | 250 | 3530 | 3530 |
| Within model R-squared | 0.587 | 0.735 | 0.488 | 0.611 | 0.266 | 0.330 |
| Between model R-squared | 0.498 | 0.958 | 0.169 | 0.851 | 0.402 | 0.590 |
| Overall R-squared | 0.259 | 0.845 | 0.056 | 0.644 | 0.171 | 0.413 |

* $p < 0.10$,

** $p < 0.05$,

*** $p < 0.01$

Clustered standard errors in parentheses.

*Oppression*. Accordingly, one would expect that government communication about vaccinations ought to focus on these two moral foundations to reduce vaccine hesitancy. We test this theory in our study. We do acknowledge that the issue of vaccine hesitancy constitutes a multi-faceted phenomenon that is influenced by a myriad of different factors [3, 49–51]. Still, the underlying $R^2$ value may be considered high enough to draw informed implications in our context and certainly represents a promising new perspective towards reducing vaccine hesitancy.

**Main result 2**. The use of the moral foundations *Authority/Subversion* and *Liberty/ Oppression* in government communication is associated with *less vaccine hesitancy*. The use of the moral foundation *Purity/Degradation* has a *weak negative effect on vaccine hesitancy*.

We find the strongest effect on the immunisation coverage for the moral foundation *Authority/Subversion* ($\beta \geq 22.452$, $p < 0.001$). This means that a one percentage point increase in the use of this dimension in RIVM's brochures would in expectation lead to an increase in vaccination rate of more than 22 percentage points. For *Liberty/Oppression* we also find a positive effect on the immunisation coverage ($\beta \geq 3.641$, $p < 0.02$). This means that the use of

*Authority/Subversion* and *Liberty/Oppression* may decrease vaccine hesitancy. Regressions (1a), (3a) and (5a) in Table A4 in S1 Appendix present an alternative regression method for which we use the *absolute* word count for the MFT dimension loading, as opposed to the *percentage* dimension loading in Table 5. This analysis indicates an effect of about 1 percentage point per signal word for *Authority/Subversion* and just below 0.3 percentage points for *Liberty/Oppression*. This means we find a robust positive effect on the vaccination rate of about respectively 1 and 0.3 percentage points per signal word.

To put the size of our coefficients into context, consider for example the last two years from our dataset. Between these two years, the overall average dimension loading for *Authority/Subversion* across all brochures has slightly increased from 0.115 in 2018 to 0.129 in 2019. With a median national cohort size in the Netherlands of about 184,000 children and an approximated lower bound effect of 22 percentage points, this translates into an expected increase of about $((0.129 - 0.115) \cdot 0.22 \cdot 184,000 \approx)$ 567 vaccinated children between 2018 and 2019. Note that this represents only the approximated isolated effect from this dimension, which only corresponds to one of the many aspects that explain the overall change in vaccination coverage.

The moral foundation *Purity/Degradation* displays a significant positive effect on immunisation coverage ($\beta = 0.809$, $p < 0.001$) only in Regression (6). We interpret this as some evidence for a small positive effect, albeit at a lower level of robustness. In Table A4 in S1 Appendix, where this dimension has significantly positive estimates in Regressions (3a) and (5a), the estimated effect translates into an increase of about 0.1 percentage points per signal word.

**Main result 3**. The use of the moral foundation *Loyalty/Betrayal* in government communication is associated with *more vaccine hesitancy*.

Our regressions discover a fairly robust effect for *Loyalty/Betrayal* ($\beta \leq -5.848$, $p < 0.001$, except for Regression (1), with $p = 0.117$). Most interestingly, this delivers evidence for a negative effect of the use of this moral foundation on the immunisation coverage. This means that the use of *Loyalty/Betrayal* in government communication may increase vaccine hesitancy by about 6–15 percentage points for every percentage point increase of this moral foundation in the brochures. When translated into a per-word measure, Table A4 in S1 Appendix indicates an effect of about −0.7 percentage points per signal word.

**Main result 4**. We find no robust significant effect for the moral foundation *Harm/Care* in government communication on vaccine hesitancy.

Our analysis finds no robust significant effect for the use of the moral foundation *Harm/Care* on the immunisation coverage ($-2.048 \leq \beta \leq 0.687$, $p > 0.0001$). This suggests no reliably discernable effect of this moral foundation in the brochures. We cannot draw a reliable conclusion on its effect on the immunisation coverage and vaccine hesitancy.

As the dimension *Fairness/Cheating* has never been used in RIVM's brochures, this factor is not included in our analysis. Hence, we cannot assess this dimension's effect on vaccine hesitancy, unfortunately.

Table 6 summarises our main results for each MFT dimension. We apply the same order as in Table 3, i.e. from most to least frequently used dimension. Most interestingly, the two dimensions that have been used most prominently in vaccination-related communication in the Netherlands appear to have either no effect on vaccination levels (Harm/Care) or only a minor, non-robust positive effect (Purity/Degradation).

**Table 6. Summary of the findings.**

| MFT Dimension | Effect of the Moral Foundation on vaccine hesitancy |
|---|---|
| Harm/Care | No robust effect on vaccine hesitancy |
| Purity/Degradation | Weak evidence for a small positive effect |
| Liberty/Oppression | Decreases vaccine hesitancy |
| Loyalty/Betrayal | Increases vaccine hesitancy |
| Authority/Subversion | Decreases vaccine hesitancy |
| Fairness/Cheating | No data |

## Discussion and conclusion

In this study, we investigate whether the use of moral foundations in government communication translates into a measurable effect on the parental choice to vaccinate their child. For this we use the Moral Foundations Dictionary [23, 29] to analyse the moral foundation loading of the vaccination information brochures from the Dutch National Institute for Public Health and the Environment (RIVM) between 2011–2019 and connect the resulting moral foundation loading with the electronic national immunisation register.

We find robust evidence for a positive relationship between the use of the moral foundations *Authority/Subversion* and *Liberty/Oppression* in government communication and vaccination uptake. Additionally, our analysis finds plausible evidence for a weak positive effect on vaccination uptake by the use of *Purity/Degradation*. Prior research has identified the MFT dimensions *Purity/Degradation* and *Liberty/Oppression* as most important for both medium and highly vaccine hesitant parents [28]. This raises the question of why we find such a strong effect on vaccination uptake by the dimension *Authority/Subversion*, if vaccine hesitant people do not specifically rely on this moral foundation? Although our study cannot provide a definitive answer and more research is needed to investigate this matter, a possible explanation may be attributed to the fact that *Authority/Subversion* has some resemblance with *Liberty/Oppression*. The moral foundation *Authority/Subversion* concerns individuals' handling of dominance and forcing beneficial relationships in hierarchies and navigating/altering behaviour in hierarchies [22, 52]. This moral foundation is currently triggered by leaders and modern institutions such as legal courts and law enforcement. The moral foundation *Liberty/Oppression* concerns individuals' autonomy and control over their own matters to keep tyrants, bullies and alpha males from becoming too powerful [22]. Future research may broaden our understanding of *why* these moral foundations in particular reduce vaccine hesitancy—and why the others do not. A somewhat speculative interpretation could be the following: imagine someone who is concerned about oppression by the government or big pharmaceutical companies and believes that these parties force vaccinations upon the population for their own interest. It is easy to imagine that this individual is more easily influenced by a sentence that appeals to the moral foundation *Liberty/Oppression*, like "*Vaccinating your children is your own choice. The government cannot force you to do so*" than by a sentence that emphasises that vaccinations do not cause much physical harm (which would appeal to the Harm/Care foundation).

While Haidt [22, Chapter 8] describes the dimension *Liberty/Oppression* as operating in tension with *Authority/Subversion*, there are clear parallels between the two. Both have a strong focus on the way in which an individual relates to institutions or persons who have a position of power/dominance towards the individual. Both were, and still are, triggered by a sort of (attempted) dominance (see, i.e. [53, 54] for a discussion on liberalism and the other moral intuitions). The relationship of these two moral foundations may explain why specifically

these two dimensions decrease vaccine hesitancy. Certainly, more research is needed to confirm or repudiate this possible explanation.

The very same "why-question" extends to the effect of *Loyalty/Betrayal*. We find a robust negative effect on vaccine uptake from usage of *Loyalty/Betrayal* in government communication. Interestingly, this is not the first time that an attempt/intervention mechanism to decrease vaccine hesitancy is found to be counterproductive. Nyhan et al. [55], for example, conduct a survey experiment in which parents receive information about the MMR vaccine. None of the applied interventions lead to an increase in vaccination intention, some even lead to a reduction and an increase in misperceptions. For future research, it may be interesting to investigate why appealing to the moral foundation *Loyalty/Betrayal* has an adverse effect on parents' vaccine hesitancy.

Translating the MFD to another language entails the possibility that remaining subtle differences in meaning may not be captured in its entirety by the translation (for a discussion of this issue for cross-cultural research, see [40, 41]). Next to this, we see three limitations of our study design. The most important limitation being the fact that we cannot control for whether parents read the brochures they were given. In fact, they might throw away the brochures before reading them. They might have also read specific parts only, which obfuscates the moral foundations that they have in fact be exposed to. However, this assumption is hard to check in any research design other than an experimental one. Future research may randomise (parts of) the brochures parents receive.

Secondly, other than official government communication, there are websites, blogs and in the Netherlands even a foundation that disseminate alternative information about childhood vaccinations (Stichting Vaccinvrij, or 'Foundation Vaccination Free,' see stichtingvaccinvrij. nl). Parents—specifically parents that may already be endowed with a proclivity to not vaccinate—may mistrust government communication (see [8, 9]) and rely on these alternative sources of information instead. The effect of this alternative information, and the use of moral foundations in this information, has not been considered in this article. Again, the direct effect of information sources may be measured most accurately using a controlled experimental research design. However, studying the effect of *alternative* information sources takes a different approach than studying the effect of using moral foundations in the context of *government communication*, which is the focus of our study.

Thirdly, events might have occurred that have had an influence on parents' vaccine hesitancy. Some examples of these events are outbreaks of diseases, an increase in reports on (alleged) adverse effects following immunisation and bad publicity in the media. The effects of these kind of events have not been taken into account in this study. We argue that these types of events should asymptotically cancel out in a wide data-set as the one we employ. Alternative approaches could be to extend the time span of the study, or considering multiple cases other than the Netherlands. In this sense, our study constitutes a part of the building block towards a better general understanding of the use of MFT dimensions in government communication.

Our results show that in the period between 2011 and 2019, the Dutch government has mainly used the moral foundations *Harm/Care* and *Purity/Degradation* in their communication towards parents, with *Liberty/Oppression* in third place. We find that the two most frequently used moral foundations have in fact no or only a minor effect on vaccination uptake. Hence, an obvious policy recommendation would be to reduce the use of these moral foundations, if government wants to optimise the effectiveness of its communication. The Dutch government, as well as other governments, should instead employ formulations triggering the moral foundations *Authority/Subversion* and *Liberty/Oppression*. Examples are readily available in the Moral Foundations Dictionary and can be translated to whichever language of choice (see for example Matsuo et al. [56]). Examples of words appealing to *Authority/*

*Subversion* are obey, duty, respect, control, refuse and oppose. Examples of words appealing to *Liberty/Oppression* are freedom, choice, right, decision and force.

In the same vein, governments should try and avoid the use of *Loyalty/Betrayal*, as this moral foundation may increase vaccine hesitancy. This means that governments should avoid the use of words such as united, community, solidarity, enemy and betrayal. Concerning the use of the moral foundation *Authority/Subversion* by the Dutch government, in particular, there is room for improvement. While our results show that this dimension has the biggest effect on reducing vaccine hesitancy, it has been one of the least-used MFT dimensions. Lastly, the effect of the use of the moral foundation *Fairness/Cheating* could not be determined as this moral foundation was never used. It may be interesting for governments to experiment with the use of this moral foundation and to track the effects on parents' vaccine hesitancy.

Our results may be of interest for the current communication on the COVID-19 vaccine. Of course, the usual caveat applies for extrapolating results into a different context. Still, many of the characteristics of our study also apply for the context of COVID-19 vaccination acceptance. What is equal is that we study communication from a public authority (a government organisation) directed at adult citizens on the topic of accepting a non-compulsory vaccination. The two most obvious differences between the COVID-19 context and the childhood vaccinations from the Dutch NIP are certainly the urgency from being in a dreadful pandemic situation and the decision to accept a vaccination for oneself versus accepting vaccinations for one's children.

Equally, the relevance of our results extends beyond our country of investigation. As before, a degree of caution applies when extrapolating results beyond the context of investigation, especially towards environments that are (culturally and politically) very different. The Netherlands is a developed democratic country with a comparatively high vaccination rate. General health care metrics (i.e. health care expenditures in % of GDP, number of physicians per 10,000 individuals, etc.) of the Netherlands are comparable to other OECD countries [57]. In terms of the Dutch culture, the Hofstede 6D model [58, 59] characterises the Netherlands as low in power distance and masculinity, average to high on uncertainty avoidance, high on indulgence, very high on individualism, and have a "pragmatic orientation" in the dimension of long term orientation. Betsch et al. [60] conduct a cross-cultural experiment on vaccination acceptance in the context of herd immunity. They find that countries that score higher on Hofstede's individualism dimension have a lower vaccination intention, but a higher responsiveness to communication on the topic of herd immunity. The influence of cultural dimensions on vaccination acceptance or vaccination communication appears to be an understudied branch of literature, which may constitute a promising avenue for future research.

Insights from this study could also help target government communication towards specific groups in the population. For example, Graham et al. [23] show that liberals and conservatives rely on different moral foundations. Tailoring the language of government communication towards its target audience may significantly improve the efficacy at which the intended message gets conveyed. Future research could involve tailoring brochures, or other types of government communication towards certain demographics and assess the effect of this measure.

## List of abbreviations (See Fig 1 and S1 Table for an overview of abbreviations in the context of the Dutch National Immunisation Programme)

LIWC Linguistic Inquiry Word Count Program
MFD Moral Foundations Dictionary [23, 29]
MFT Moral Foundations Theory

NIP        National Immunisation Programme

RIVM     Dutch National Institute for Public Health and the Environment

## Supporting information

**S1 Appendix. Discussion on Moral Foundations Theory and results from an analysis using absolute effects.**
(ZIP)

**S1 Table. Overview immunisation coverage.** Immunisation coverage in the Netherlands for the vaccinations of the National Immunisation Programme.
(ZIP)

**S1 File. Dutch Moral Foundations Dictionary.** Translation of the original Moral Foundations Dictionary.
(XLSX)

## Acknowledgments

We thank the Dutch National Institute for Public Health and the Environment (RIVM) for their invaluable support in collecting the required brochures.

## Author Contributions

**Conceptualization:** Florian Heine, Ennie Wolters.

**Data curation:** Florian Heine, Ennie Wolters.

**Formal analysis:** Florian Heine, Ennie Wolters.

**Investigation:** Florian Heine, Ennie Wolters.

**Methodology:** Florian Heine, Ennie Wolters.

**Project administration:** Florian Heine.

**Resources:** Florian Heine, Ennie Wolters.

**Software:** Florian Heine, Ennie Wolters.

**Supervision:** Florian Heine.

**Validation:** Florian Heine, Ennie Wolters.

**Visualization:** Florian Heine.

**Writing – original draft:** Ennie Wolters.

**Writing – review & editing:** Florian Heine.

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
