## [Decision Letter · Decision Letter 0]

8 Jun 2021

PONE-D-21-07601

Using moral foundations in government communication to reduce vaccine hesitancy

PLOS ONE

Dear Dr. Heine,

Thank you for submitting your manuscript to PLOS ONE. After careful consideration, we feel that it has merit but does not fully meet PLOS ONE’s publication criteria as it currently stands. Therefore, we invite you to submit a revised version of the manuscript that addresses the points raised during the review process.

Both reviewers provided important comments (see below). Please read the comments and incorporate those in the revision.

We look forward to receiving your revised manuscript.

Kind regards,

Kazutoshi Sasahara

Academic Editor

PLOS ONE

Journal Requirements:

Reviewers' comments:

Reviewer's Responses to Questions

**Comments to the Author**

1. Is the manuscript technically sound, and do the data support the conclusions?

Reviewer #1: Partly

Reviewer #2: Yes

2. Has the statistical analysis been performed appropriately and rigorously? 

Reviewer #1: Yes

Reviewer #2: Yes

3. Have the authors made all data underlying the findings in their manuscript fully available?

Reviewer #1: Yes

Reviewer #2: No

4. Is the manuscript presented in an intelligible fashion and written in standard English?

Reviewer #1: Yes

Reviewer #2: No

5. Review Comments to the Author

Reviewer #1: GENERAL COMMENTS:

*Introduction section*

What is “intentions”?

The authors mention the term “intentions” on page 2 as the concept that has been focused on previous studies, but it is not clearly stated what is it and why it is not suitable for this kind of research. I assume that the concept of “intensions” represents one’s self-stated vaccine hesitancy from the sentences in the same paragraph, but I am not sure.

Explanation of the MFT. The explanation of the MFT can be included in the manuscript itself (not in Appendix) because the flow of the manuscript should not be interrupted. I am not sure why the authors explain it in the separated section.

Relationship between concepts

One of my major concerns is that how morality and vaccine hesitancy are related is not clearly discussed in the Introduction section, which is connected to the aim of this study. The authors describe the MFT and the past research on the MFT and vaccine uptake, but, in the first place, how morality itself and vaccine uptake/hesitancy are related and why the authors should conduct research on vaccine hesitancy with the framework of morality are not argued. The authors may need to explain morality before the MFT to clarify the importance of their study.

Purpose of the study

The last paragraph on page 3 seems to describe the purpose of the present study, but what will be revealed by this study and the importance of this study in the academic field are not elaborated enough. Particularly this paragraph confused me because all the sentences are written in present tense, and also because what will be investigated and the results are presented in the same paragraph. Further, the authors need hypothesis; or, isn’t it a hypothesis-testing type study?

How the purpose is related to the paradigm

Related to the purpose of the study, the authors need to justify the appropriateness of the analyses to accomplish what they want to know. Why do the authors look at the language use? I would like to know more about the rationale.

*Methods section*

I like to see some example words from the MFD. Also, the Dutch version of the MFD needs to be described so that a reader understands it (e.g., How was the original version translated? Is the translation method a prevalent way to be used in past research involving non-English speaking people?).

*Results section*

The authors should present their results more carefully from regression analyses, such as “A significantly predicted B.” The expression used by the authors (e.g., “leads to”) do not efficiently convey what was observed. Also, it is not clear why the authors cited Amin et al. in the Main Result 1 section because Amin et al. found the reverse causal relationship compared to the current study. Please let me know if I am reading wrong. Regarding Main Result 4, I am not sure why the authors mention the Harm/Care foundation specifically. I like to see the rationale in the introduction section.

Results and interpretations

I see the results from the analyses and their interpretations are sometimes mixed up both in style- and concept-wise. For example, the descriptions in lines 93-96 should be in the results section. For another example, although you state that immunization coverage is inversely related to vaccine hesitancy (lines 187-188), it does not automatically mean that “the use of moral foundations in government communication affects vaccine hesitancy (lines 322-323).” What was found in the regression analyses was that the use of moral-related language significantly predicted the vaccine rate. I sometimes have hard time following the authors’ logic—what was FOUND as results and what was SUGGESTED from the results? Further, when the regression is performed, the authors may want to clearly state the independent variables and dependent variables.

Minor issue1: I am not sure if your style is fine in your field, but Methods and Results should be basically written in past tense.

Minor issue 2: I am not sure how the authors use the term “effect size” (e.g., line 348). I was wondering if they calculated Cohens’ d etc.

*Discussion section*

The authors state “future research may investigate the question of why these moral foundations in particular reduce vaccine hesitancy,” but isn’t it the purpose of the current study? It is related to the purpose of this study—the authors need to justify why this particular study was designed this way. It is not just that the variables were randomly chosen or the authors unexpectedly found the associations between them. Further, the authors discuss the possible practical applications of the results from this study, but I like to see how this study can contribute theoretically.

Minor issue: It is easier to understand if the discussion section starts with the summary of the study and results.

*Other*

Tone of the manuscript

The tone of the manuscript (especially results and discussion) seems a bit assertive. I’d caution against the causal language used in some sentences throughout the manuscript, such as “…as this moral foundation increases vaccine hesitancy…(line 473-). Again, it was just that the use of morality-related language predicted/was associated with vaccine rate (positively or negatively).

What the “use of moral foundations” means

The other of my major concerns is the language used in the manuscript. The meaning of the expression, “use of moral foundations,” sounds vague. It is the frequency of the morality-related word. I like to see how the language use and vaccine ratio are conceptually connected.

SPECIFIC COMMENTS:

Page 11: I like to see how N = 22 was calculated; I like to see the number of the broachers for each year.

Tables: The decimals should be aligned.

Page 20, Line 429: Please elaborate how it is counterproductive.

Page 20, Line 451: What is the “underlying study”?

Page 21, Line 469: Please provide some examples of the MFD in different languages.

Appendix (Page 233, Line 528): The description of the pluralist approach is confusing. Does pluralism demur the evolutionary explanation?

Reviewer #2: The paper studies the relationship between the moral dimensions and the parental hesitancy towards vaccination. It is a relevant topic given the current pandemic situation. Few comments are as follows:

Major comments

1. Please describe the process used to translate the English MFD dictionary to Dutch. Moral foundations may vary with culture and usage of the English-translated dictionary may not be able to capture the socio-cultural aspects of a particular region and may potentially bias the results. How do you ensure that English dictionary can be generalised well to the Dutch language? Please refer to [1] to learn more about the translation methodology.

2. The authors have used a moral foundation dictionary (MFD) and word count based methodology to capture the moral rhetoric used in the brochures. MFD created by a small group of experts with limited amounts of stem words and lemmas may not be valid for diverse contexts and populations [2]. The authors are advised to look at the applicability of the extended MFD [2] for their work. Also, many machine learning and word embedding based methods have been shown to better capture the moral rhetoric than word count based method. Authors should consider using such methods ( for example [3,4]) .

3. As mentioned by the authors the NIP started in 1957, please mention the reasons to select the data from 2011-2019 only.

4. In equation (1), moral dimensions i.e. HC, PD etc have been shown as varying with regions. Are there different brochures corresponding to the different regions? Kindly clarify. It seems there are only 22 brochures in total from 2011 to 2019. The immunisation coverage data is available at the high spatial granularity, while the same is not true for moral dimensions data. How do you ensure that the OLS model is not overfitting the data?

5. How do you account for the different lengths of brochures in Table 5? What was the motivation to use absolute word count?

6. Fairness dimension was not detected in the brochures and it is also the most difficult dimension to trace in language [5]. Authors may include this.

Minor comments

7. Kindly include some example statements from the vaccination brochures corresponding to different moral dimensions.

8 . The overall writing can be improved significantly. Few examples

(a)Page 4, line 73, 76, Subsection has been mentioned without referring to one.

(b) Page 16, line 330, “We test this theorem in our study”. Usage of the word “theorem” is incorrect.

(c) Please avoid vague terms such as “some” for results where statistical data can be provided e.g. Table 6 ”some evidence”

9. Table 10 has been referred to in the text after Table 2. Please put them in a sequential order.

10. n Figure 2, for better readability, please include 2012, 2014, etc on x-axis. Also, a descriptive caption can be provided which may help in better understanding of the figure.

[1] Matsuo, Akiko, et al. "Development and validation of the japanese moral foundations dictionary." PloS one 14.3 (2019): e0213343.

[2] Hopp, Frederic R., et al. "The extended Moral Foundations Dictionary (eMFD): Development and applications of a crowd-sourced approach to extracting moral intuitions from text." Behavior Research Methods 53.1 (2021): 232-246.

[3] Sagi, Eyal, and Morteza Dehghani. "Measuring moral rhetoric in text." Social science computer review 32.2 (2014): 132-144

[4] Araque, Oscar, Lorenzo Gatti, and Kyriaki Kalimeri. "MoralStrength: Exploiting a moral lexicon and embedding similarity for moral foundations prediction." Knowledge-based systems 191 (2020): 105184

[5] Kennedy, Brendan, et al. "Moral concerns are differentially observable in language." Cognition 212 (2021): 104696

6. PLOS authors have the option to publish the peer review history of their article (what does this mean?). If published, this will include your full peer review and any attached files.

Reviewer #1: No

Reviewer #2: No

---

## [Author Response · Author response to Decision Letter 0]

10 Aug 2021

Reviewer 1

Thank you for your helpful feedback on our manuscript! Please find in the following a brief overview of how we addressed each of your comments.

Comment 1 *Introduction section*

What is “intentions”?

The authors mention the term “intentions” on page 2 as the concept that has been focused on previous studies, but it is not clearly stated what is it and why it is not suitable for this kind of research. I assume that the concept of “intensions” represents one’s self-stated vaccine hesitancy from the sentences in the same paragraph, but I am not sure.

Response 1

We add a sentence to clarify this on page 2. Indeed “intentions” refers to the self-stated intention to get vaccinated. As some individuals may end up not getting a vaccination despite their stated intention, this indication is somewhat noisy. Accordingly, we make the argument that governmental records, by contrast, promise to deliver a more accurate measure [1].

Comment 2 Explanation of the MFT.

The explanation of the MFT can be included in the manuscript itself (not in Appendix) because the flow of the manuscript should not be interrupted. I am not sure why the authors explain it in the separated section.

Response 2

We would suppose that adding the fairly elaborated explanation of MFT in the Introduction may unnecessarily bloat it to a point that it loses focus. We believe there are three alternatives: 1) Introducing MFT briefly in the Introduction and referring the reader to a detailed appendix (this is the alternative we have opted for). 2) Transferring the text from

the appendix into the introduction. This would make the introduction about seven pages long, which does not seem to be a desirable option. 3) Taking some more of the content from the Appendix into the Introduction. This options seems more viable than option 2, yet it may create a situation in which neither the appendix nor the somewhat more elaborate explanation in the Introduction remain as a solid self-sustained piece that adds value to the text. We respectfully believe that readers who are not at all familiar with MFT would profit from a thorough introduction of the concept, as presented in the appendix, while readers who do know MFT would not be served with a lengthy overview of the concept of MFT in the introduction.

Comment 3 Relationship between concepts

One of my major concerns is that how morality and vaccine hesitancy are related is not clearly discussed in the Introduction section, which is connected to the aim of this study. The authors describe the MFT and the past research on the MFT and vaccine uptake, but, in the first place, how morality itself and vaccine uptake/hesitancy are related and why the authors should conduct research on vaccine hesitancy with the framework of morality are not argued. The authors may need to explain morality before the MFT to clarify the importance of their study.

Response 3

Thank you for pointing out the lack of explicit link between morality/intuitions and vaccine hesitancy in our introduction. We now make this more explicit towards the end of page 2.

Comment 4 Purpose of the study

The last paragraph on page 3 seems to describe the purpose of the present study, but what will be revealed by this study and the importance of this study in the academic field are not elaborated enough. Particularly this paragraph confused me because all the sentences are written in present tense, and also because what will be investigated and the results are presented in the same paragraph. Further, the authors need hypothesis; or, isn’t it a hypothesis-testing type study?

Response 4

The paragraph starting “We use the...” provides a very brief outline of our method and our results. In the paragraph before (starting “Amin et. al...”) we establish our article’s contribution to the literature and which knowledge gap we address. In particular, our study is the first to connect the MFT dimensions used in vaccine related government communication towards parents with actual governmental care records on vaccination uptake. By doing so we explore whether triggering specific moral foundations translates into measurable change in behaviour and vaccination uptake. As for the exploratory character of our study we decided not to work with pre-formulated hypotheses other than the benchmarks derived from Amin et al.’s study.

Comment 5 How the purpose is related to the paradigm Related to the purpose of the study, the authors need to justify the appropriateness of the analyses to accomplish what they want to know. Why do the authors look at the language

use? I would like to know more about the rationale.

Response 5

We hope that the elaborations added to the Introduction in the context discussed above have clarified the link between the study’s purpose and the applied methods. In short, prior research suggests that vaccine hesitancy may be rooted in fundamental intuitions and emotions. MFT was developed to identify these intuitive ethics that guide people’s behaviour. To measure MFT, researchers usually look at some form of communication and then assess the language employed to identify which dimensions are triggered more prominently. Our study positions itself in this tradition, extending the context towards revealed behaviour and vaccination acceptance.

Comment 6 *Methods section*

I like to see some example words from the MFD. Also, the Dutch version of the MFD needs to be described so that a reader understands it (e.g., How was the original version translated? Is the translation method a prevalent way to be used in past research involving non-English speaking people?).

Response 6

Thank you for pointing out that the original manuscript has been too brief on this aspect. We now add the following explanation on page 9: “Similar to the method outlined by Harzing [2], both authors who are fluent/native Dutch speakers, independently translate the original English MFD words and word stems into Dutch and resolve any differences in the separately generated translations by discussion.” We include the Dutch MFD as supporting information, which will also become available to the reader in the Online Appendix. On page 29 we add a table with example sentences in their original and the translated form for each dimension. On page 9 we add a reference to the table in the main text. Further, we include some example words from the MFD on page 21 and an example sentence on page 16. The complete MFD can be accessed via https://moralfoundations.org/wp-content/ uploads/files/downloads/moral%20foundations%20dictionary.dic.

Comment 7 *Results section*

The authors should present their results more carefully from regression analyses, such as “A significantly predicted B.” The expression used by the authors (e.g., “leads to”) do not efficiently convey what was observed. Also, it is not clear why the authors cited Amin et al. in the Main Result 1 section because Amin et al. found the reverse causal relationship compared to the current study. Please let me know if I am reading wrong. Regarding Main Result 4, I am not sure why the authors mention the Harm/Care foundation specifically. I like to see the rationale in the introduction section.

Minor issue1: I am not sure if your style is fine in your field, but Methods and Results should be basically written in past tense.

Minor issue 2: I am not sure how the authors use the term “effect size” (e.g., line 348). I was wondering if they calculated Cohens’ d etc.

Response 7

Thank you for pointing this out. We rephrased the main results towards more careful language. The study by Amin et al. is most closely related to our study. Accordingly we find a comparison between their results and ours most insightful. In our view, the fact that some of our results show a different directionality than in Amin et al. deserves considerable attention in our article. We think the explicit reference to Amin et al.’s results is in the interest of transparency here. We discuss all MFT dimensions in the main results. The fact that we find no significant effect for a particular MFT dimension conveys an interesting insight too. In a well-powered study, ignoring non-significant results in the body of scientific evidence would feed the publication bias.

In our field it is common to employ present tense throughout the article, except when the context explicitly calls for pas tense, like for example “At one point, Keynesian (1932) logic was predominant, but theories beginning with Lucas overthrew it.” If the editor or journal has a specific preference, we are happy to revise this at an eventual typesetting phase. In the particular OLS fixed effects regression we employ, we can simply use the regression coefficient to make a statement about the size of the underlying effect. To avoid confusion with Cohen’s f2, which is often used to measure the overall effect size for a multiple regression, we apply reformulations where applicable and make sure that we are talking about the effect of A on B, not the overall effect size for the multiple regression.

Comment 8 *Discussion section*

The authors state “future research may investigate the question of why these moral foundations in particular reduce vaccine hesitancy,” but isn’t it the purpose of the current study? It is related to the purpose of this study—the authors need to justify why this particular study was designed this way. It is not just that the variables were randomly chosen or the authors unexpectedly found the associations between them. Further, the authors discuss the possible practical applications of the results from this study, but I like to see how this study can contribute theoretically.

Minor issue: It is easier to understand if the discussion section starts with the summary of the study and results.

Response 8

Thank you for pointing this out to us. The formulation we have chosen here may have been a bit too modest. It is not that we offer no explanation for the effects found through our analysis. In the discussion we offer an interpretation of which channels contribute to the observed effect and our interpretation of the underlying driving force. We change the formulation on page 19 to properly reflect that.

In the Introduction (specifically on page 3), we formulate the contribution of our study to the scientific debate. Reiterated very briefly, this would be to investigate whether the use of moral foundations in government communication translates into measurable change in behaviour, here specifically in the context of vaccination acceptance. Theoretical or empirical studies focussing on explaining the social and psychological chain of causality, however, would require a research design that allows zooming in on the motivational channels to accept a vaccination. This could be achieved through a survey design or the use of carefully selected micro-type (i.e. census) data.

Thank you for your suggestion to open the Discussion and Conclusion with a brief summary of the study and its purpose. We now add a brief paragraph to accommodate this.

Comment 9 *Other*

Tone of the manuscript

The tone of the manuscript (especially results and discussion) seems a bit assertive. I’d caution against the causal language used in some sentences throughout the manuscript, such as “...as this moral foundation increases vaccine hesitancy...” (line 473-). Again, it was just that the use of morality-related language predicted/was associated with vaccine rate (positively or negatively).

Response 9

Thank you for pointing this out to us. We have revised the tone of many of the formulations in this vein, toning down the assertive character of the language in the Results and Discussion Sections.

Comment 10 What the “use of moral foundations” means

The other of my major concerns is the language used in the manuscript. The meaning of the expression, “use of moral foundations,” sounds vague. It is the frequency of the moralityrelated word. I like to see how the language use and vaccine ratio are conceptually connected.

Response 10

Indeed “use of moral foundations” refers to the frequency of morality-related words in a given text, but also to which specific dimension gets triggered. If referred to “the frequency of morality-related words” instead, we would give the false impression that we would only look at a morality-related “bag of words” and sell short the detail of our analysis. We hope that the explicit link between intuition and vaccine acceptance on page 2 creates a more coherent picture of the conceptual connection between moral language and vaccine hesitancy.

Comment 11 SPECIFIC COMMENTS:

Page 11: I like to see how N = 22 was calculated; I like to see the number of the broachers

for each year.

Tables: The decimals should be aligned.

Page 20, Line 429: Please elaborate how it is counterproductive.

Page 20, Line 451: What is the “underlying study”?

Page 21, Line 469: Please provide some examples of the MFD in different languages. Response 11 For the information on a precise breakdown of brochures per year we kindly refer to Table 2. All vaccination brochures and the respective year of publication are listed there. We fixed the decimal alignment in Table 4. We add some brief elaboration on the point of some interventions turning out counterproductive. The underlying study is our study. We change the wording accordingly to avoid confusion. Lastly, we provide an example for a translation of the MFD into Japanese by Matsuo et al. [3].

Comment 12 Appendix (Page 233, Line 528): The description of the pluralist approach is confusing. Does pluralism demur the evolutionary explanation?

Response 12

Thank you for discovering this mistake. The term “demur” is incorrect here. The intention of the sentence on line 528 was, in other words, to express that pluralists object the monist perspective and argue that “evolutionary thinking encourages pluralism”. We replace the term “demur” by “argue”.

Reviewer 2

Thank you for your helpful feedback on our manuscript! Please find in the following a brief overview of how we addressed each of your comments.

Comment 1 Please describe the process used to translate the English MFD dictionary to Dutch. Moral foundations may vary with culture and usage of the English-translated dictionary may not be able to capture the socio-cultural aspects of a particular region and may potentially bias the results. How do you ensure that English dictionary can be generalised well to the Dutch language? Please refer to [3] to learn more about the translation methodology.

Response 1

Thank you for pointing out that the original manuscript has been too brief on this aspect. We now add the following explanation on page 9: “Similar to the method outlined by Harzing [2], both authors who are fluent/native Dutch speakers, independently translate the original English MFD words and word stems into Dutch and resolve any differences in the separately generated translations by discussion.”

In contrast to our translation method, Matsuo et al. [3] transform the English word stems into a subset of words. More concretely, we understand that for each word stem (they employ the example “justifi*”), Matsuo et al. [3] use a website to find the 11 most common words using a particular word stem (here: “justifiable”, “justifiableness”, etc.) and discard all other realisations. We respectfully believe that keeping the word stems allows us to cover a larger domain of words that may fall into the category. By the nature of Matsuo et al. [3]’s technique, by contrast, a certain number of potential words do not make the cut for the 11 most common words for a given word stem. The linguistic distance between English and Japanese probably requires this approach (as Japanese, for example, lacks the concept of punctuating words by blank spaces), but in our case we remain in the same language family with a very similar word and sentence structure.

Another difference between the approach by Matsuo et al. [3] and our study is that while we translate the words and word stems by human translators, Matsuo et al. [3] use an online dictionary and web scraping. We think both methods are theoretically feasible and valid.

Comment 2 The authors have used a moral foundation dictionary (MFD) and word count based methodology to capture the moral rhetoric used in the brochures. MFD created by a small group of experts with limited amounts of stem words and lemmas may not be valid for diverse contexts and populations [4]. The authors are advised to look at the applicability of the extended MFD [4] for their work. Also, many machine learning and word embedding based methods have been shown to better capture the moral rhetoric than word count based method. Authors should consider using such methods (for example [5, 6]).

Response 2

The eMFD [4], published last month, is certainly a promising new tool for research on MFT. The method we employ in our study, i.e. the MFD-based word count method, has established itself among a long stream of literature, i.e. [7, 8, 9, 10]. We believe the eMFD has opened a promising avenue towards an alternative approach for analysing the use of moral language in text. However, considering that the eMFD contains a different list of words than the MFD, the novelty of the eMFD and the established character of our text analysis method, we believe that utilising the eMFD for our research question would be a different paper and lies beyond the scope of this study.

Newly developed machine learning and word embedding methods have been described as superior to the keyword-based methods like LIWC when measuring the rhetoric related to specific topics in a text. For example, when analysing a text on the war on drugs, a given text may also discuss money laundering or other criminal activities. Methods as developed by Sagi and Dehghani [5] are able to measure the moral dimensions of each topic, rather than the text as a whole. Sagi and Dehghani [5] argue: “While keyword-based methods like LIWC can be used to measure the moral rhetoric over entire documents, we are interested in measuring the rhetoric related to specific topics.” The purpose of our study, however, is to analyse the moral rhetoric in government-issued vaccination brochures. These brochures are of interest to us in their entirety, for why we think it is safe to utilise the LIWC word count based method in our study.

On a different note, both the eMFD, as well as the machine learning approaches are currently limited to the five original 

 MFT dimensions and do not cover the sixth dimension Liberty/Oppression. In our study we are able to also include the sixth dimension into the analysis.

Comment 3 As mentioned by the authors the NIP started in 1957, please mention the reasons to select the data from 2011-2019 only.

Response 3

RIVM has started using standardised brochures to parents in 2011.

Comment 4 In equation (1), moral dimensions i.e. HC, PD etc have been shown as varying with regions. Are there different brochures corresponding to the different regions? Kindly clarify. It seems there are only 22 brochures in total from 2011 to 2019. The immunisation coverage data is available at the high spatial granularity, while the same is not true for moral dimensions data. How do you ensure that the OLS model is not overfitting the data?

Response 4

Thank you for noticing this notation mistake and apologies for the confusion this may have caused! Indeed, there are no specific brochures per region, so the “j”-indices are incorrect for the MFT dimensions in Equation (1). We will drop them from the manuscript. Concerning the issue of overfitting, we understand this problem occurs when a regression has too many dependent variables compared to the number of observations. To our knowledge, the inverse (a comparably small amount of dependent variables to describe a rich set of independent variables) does not constitute an overfitting problem. We believe the issue of dummy variables to be a good counter-example, in which only the absence or presence of a phenomenon is expressed on the part of the dependent variables. This popular regression technique finds widespread applicability in research while by its very nature the independent variables can only ever have two different values, zero or one. The MFT dimensions which we employ, by contrast, contain a much richer domain and bring about ample sufficient variation to the set of dependent variables.

Comment 5 How do you account for the different lengths of brochures in Table 5? What was the motivation to use absolute word count?

Response 5

Thank you for suggesting to control for the length of brochures. We have added the absolute word count per brochure as additional fourth control in Regressions (2), (4) and (6). The directionality of the results remains largely unchanged. We update the discussion of the results accordingly. The motivation for also conducting an analysis using an absolute word count measure in Table 9 was to provide a more transparent and accessible idea of the actual impact of our results. From a public health perspective, percent differences can be small but meaningful across a population. The absolute word count allows us to interpret the magnitude of our estimates as effect per morally relevant signal word.

Comment 6 Fairness dimension was not detected in the brochures and it is also the most difficult dimension to trace in language [11]. Authors may include this. 

Response 6

Thank you for pointing out this interesting study. We agree that this finding is very relevant for us in the context of the traceability of Fairness/Cheating and we gladly add a reference to it on page 12 of the manuscript.

Comment 7 Kindly include some example statements from the vaccination brochures corresponding to different moral dimensions.

Response 7

Thank you for this idea which helps make our study more transparent and accessible. On page 29 we add a table with example sentences in their original and the translated form for each dimension. On page 9 we add a reference to the table in the main text.

Comment 8 The overall writing can be improved significantly. Few examples

(a) Page 4, line 73, 76, Subsection has been mentioned without referring to one.

(b) Page 16, line 330, “We test this theorem in our study”. Usage of the word “theorem” is incorrect.

(c) Please avoid vague terms such as “some” for results where statistical data can be provided e.g. Table 6 “some evidence”

Response 8

Thank you for pointing out the missing reference link to the respective subsections on page 4. On page 16 we change the term “theorem” to “theory”. Table 6 serves as a rough overview of the findings. The text and the regression tables present a much more concrete quantification of the results. We adjust “some evidence” to “weak evidence” as it may better represent the general character of the results for this particular MFT dimension.

Comment 9 Table 10 has been referred to in the text after Table 2. Please put them in a sequential order.

Response 9

Table 10 is in the appendix while Table 2 is part of the main text. As by the journal typesetting guidelines, the numbering of the tables is in the order at which they appear in the text, followed by tables that appear in the appendix. If the editor or journal has a specific preference, we are happy to revise this at an eventual typesetting phase.

Comment 10 n Figure 2, for better readability, please include 2012, 2014, etc on x-axis. Also, a descriptive caption can be provided which may help in better understanding of the figure.

Response 10

We added the years in the graph. Please find the caption for Figure 2 on page 11. PLOS ONE typesetting guidelines require authors to put figures after the manuscript and without caption. The caption is included in the manuscript text at the position where the figure would eventually be.

References

1. Rodewald L, Maes E, Stevenson J, Lyons B, Stokley S, Szilagyi P. Immunization performance measurement in a changing immunization environment. Pediatrics. 1999;103(Supplement 1):889–897.

2. Harzing AW. Does the use of English-language questionnaires in cross-national research obscure national differences? International Journal of Cross Cultural Management. 2005;5(2):213–224.

3. Matsuo A, Sasahara K, Taguchi Y, Karasawa M. Development and validation of the Japanese Moral Foundations Dictionary. PLOS ONE. 2019;14(3):e0213343.

4. Hopp FR, Fisher JT, Cornell D, Huskey R,Weber R. The extended Moral Foundations Dictionary (eMFD): Development and applications of a crowd-sourced approach to extracting moral intuitions from text. Behavior Research Methods. 2021;53(1):232–246.

5. Sagi E, Dehghani M. Measuring moral rhetoric in text. Social Science Computer Review. 2014;32(2):132–144.

6. Araque O, Gatti L, Kalimeri K. MoralStrength: Exploiting a moral lexicon and embedding similarity for moral foundations prediction. Knowledge-Based Systems. 2020;191:105184.

7. Long JA, Eveland Jr WP. Entertainment Use and Political Ideology: Linking Worldviews to Media Content. Communication Research. 2021;48(4):479–500.

8. Wheeler MA, McGrath MJ, Haslam N. Twentieth century morality: The rise and fall of moral concepts from 1900 to 2007. PLoS one. 2019;14(2):e0212267.

9. Mooijman M, Hoover J, Lin Y, Ji H, Dehghani M. Moralization in social networks and the emergence of violence during protests. Nature human behaviour. 2018;2(6):389– 396.

10. Clifford S, Jerit J. How words do the work of politics: Moral foundations theory and the debate over stem cell research. The Journal of Politics. 2013;75(3):659–671.

11. Kennedy B, Atari M, Davani AM, Hoover J, Omrani A, Graham J, et al. Moral concerns are differentially observable in language. Cognition. 2021;212:104696.

---

## [Decision Letter · Decision Letter 1]

15 Sep 2021

PONE-D-21-07601R1Using moral foundations in government communication to reduce vaccine hesitancyPLOS ONE

Dear Dr. Heine,

Thank you for submitting your manuscript to PLOS ONE. After careful consideration, we feel that it has merit but does not fully meet PLOS ONE’s publication criteria as it currently stands. Therefore, we invite you to submit a revised version of the manuscript that addresses the points raised during the review process.

We look forward to receiving your revised manuscript.

Kind regards,

Kazutoshi Sasahara

Academic Editor

PLOS ONE

Journal Requirements:

Additional Editor Comments (if provided):

Both reviewers agreed that the MS was improved. However, they also think that the MS needs several minor revisions. Please read their comments and address them properly.

Reviewers' comments:

Reviewer's Responses to Questions

**Comments to the Author**

1. If the authors have adequately addressed your comments raised in a previous round of review and you feel that this manuscript is now acceptable for publication, you may indicate that here to bypass the “Comments to the Author” section, enter your conflict of interest statement in the “Confidential to Editor” section, and submit your "Accept" recommendation.

Reviewer #1: All comments have been addressed

Reviewer #2: (No Response)

2. Is the manuscript technically sound, and do the data support the conclusions?

Reviewer #1: Yes

Reviewer #2: Yes

3. Has the statistical analysis been performed appropriately and rigorously? 

Reviewer #1: Yes

Reviewer #2: Yes

4. Have the authors made all data underlying the findings in their manuscript fully available?

Reviewer #1: Yes

Reviewer #2: Yes

5. Is the manuscript presented in an intelligible fashion and written in standard English?

Reviewer #1: Yes

Reviewer #2: Yes

6. Review Comments to the Author

Reviewer #1: I am happy with the revision and only have some minor suggestions/comments as follows:

-In the subsection “The Dutch National Immunisation Programme (p. 5)”, the authors state “Our results confirm this trend, providing… (L. 103)”. Is it from the current study? If so, is it appropriate to show a part of the results here in the method section?

-Now that the authors clarify how they made the Dutch version of the MFD, I suggest the authors mention the translation issue as a limitation.

-In the Discussion and Conclusion section, the authors argue a possible difference between Authority and Liberty foundations. I would like to see some articles discussing the relationships between those (or other) foundations (if any). It can help readers to deeply speculate the results from the current study and moral foundations as a whole for future research.

-Thank you for showing how manuscripts are structured in your field (Response 7). It helps me.

Reviewer #2: Table 2 lists 26 brochures. Please highlight the brochures which were not considered for analysis in the table itself to avoid the confusion.

The paper needs serious proof-reading. There are several grammatical mistakes and other errors. e.g. the reference to subsections are missing at many places : pg 17- line 338, pg 10- line 222, page 10-line 224.

As mentioned in pg 17, line 335, fairness/cheating has been discarded from analysis. FC should also be removed from Equation 1.

Since MF dimensions do not vary spatially. I do not see the point of analysing it at national, regional and municipal levels. It is rather surprising that the coeffs for MF dimensions vary significantly at the different spatial resolutions since the MF dimensions and other control variables such as degree of difficulty (which are again derived from linguistic analysis of brochures) and time remain constant across the regions. Is it only modelling the variations in population sizes? Loyalty/betrayal effect became significant at regional and municipal level but not at the national level, however, the values of MF dimensions remain the same.

OLS with time-demeaning fixed effects would suggest that the moral dimensions remain constant over time. Is this a valid assumption?

Authors should not make strong claims regarding the effect of MF dimensions on vaccine hesitancy on the basis of only 22 sample points and there could be many other factors affecting the hesitancy which have not been considered for the analysis. Also, consider that the moral foundations are not completely independent of each other which is also evident from the MF dictionary words.

For line and page numbers, please refer to manuscript with track changes

7. PLOS authors have the option to publish the peer review history of their article (what does this mean?). If published, this will include your full peer review and any attached files.

Reviewer #1: No

Reviewer #2: No

---

## [Author Response · Author response to Decision Letter 1]

11 Oct 2021

Reviewer 1

Thank you for your kind interest in our study and for the helpful feedback throughout the review process! Please find in the following a brief overview of how we addressed each of your comments.

Comment 1: In the subsection ``The Dutch National Immunisation Programme'' (p. 5), the authors state ``Our results confirm this trend, providing...'' (L. 103). Is it from the current study? If so, is it appropriate to show a part of the results here in the method section?

Response 1

We agree that this sentence is somewhat premature at this particular position in the manuscript. We present our empirical results concerning the time trend for vaccination acceptance in the Results Section on page 17, which we agree is a better position for this discussion. We propose to delete the sentence you refer to on page 5.

Comment 2

Now that the authors clarify how they made the Dutch version of the MFD, I suggest the authors mention the translation issue as a limitation.

Response 2

We now mention this as limitation both in the Methods Section and in the Discussion and Conclusion.

Comment 3

In the Discussion and Conclusion section, the authors argue a possible difference between Authority and Liberty foundations. I would like to see some articles discussing the relationships between those (or other) foundations (if any). It can help readers to deeply speculate the results from the current study and moral foundations as a whole for future research.

Response 3

Thank you for proposing this valuable addition to the manuscript, which we gladly include.

Comment 4

Thank you for showing how manuscripts are structured in your field (Response 7). It helps me.

Response 4

We are glad we could clarify this issue. We would like to reiterate that we are happy to adapt this editorial aspect of our article in an eventual typesetting phase, if the editor feels this would better fit PLOS ONE's publication culture.

Reviewer 2

Thank you for your kind interest in our study and for the helpful feedback throughout the review process! Please find in the following a brief overview of how we addressed each of your comments.

Comment 1

Table 2 lists 26 brochures. Please highlight the brochures which were not considered for analysis in the table itself to avoid the confusion.

Response 1

We now recognise how the representation of Table 2 seems to have caused confusion. Thank you for reiterating this issue to bring it to our attention again. As some brochures have comparably long names (i.e. ``Folder baby's van 2, 3, 4 en 11 maanden 2011''), there is a line break in the particular table cell. We now apply alternating light-grey row colours to tell different brochures from mere line breaks.

Comment 2

The paper needs serious proof-reading. There are several grammatical mistakes and other errors. e.g. the reference to subsections are missing at many places : pg 17- line 338, pg 10- line 222, page 10-line 224.

Response 2

We have performed another round of proofreading and adjusted a number of formulations which we hope elevates the language use within our manuscript to an acceptable level. We apologise for the dead links/references to some of the sections and subsections which we caused by changing the associated command structure within our typesetting software. We hope this issue has been resolved now.

Comment 3

As mentioned in pg 17, line 335, fairness/cheating has been discarded from analysis. FC should also be removed from Equation 1.

Response 3

Thank you for pointing this out to us. We agree that this dimension may be discarded from Equation 1. For consistency sake we also dropped it from our regression outputs in Tables 5 and 10.

Comment 4

Since MF dimensions do not vary spatially. I do not see the point of analysing it at national, regional and municipal levels. It is rather surprising that the coeffs for MF dimensions vary significantly at the different spatial resolutions since the MF dimensions and other control variables such as degree of difficulty (which are again derived from linguistic analysis of brochures) and time remain constant across the regions. Is it only modelling the variations in population sizes? Loyalty/betrayal effect became significant at regional and municipal level but not at the national level, however, the values of MF dimensions remain the same.

Response 4

The purpose of executing the analysis at different spatial levels is to assess the robustness of the results. Concretely, we are interested in demonstrating the results' sensitivity to different ways of measuring the same thing. We add a brief clarification on page 14.

Comment 5

OLS with time-demeaning fixed effects would suggest that the moral dimensions remain constant over time. Is this a valid assumption?

Response 5

The fixed effects panel is not the moral dimensions, which indeed vary over time. Our panel is the type of vaccination in a region. For data on the national level, this would be one of ten vaccinations (i.e. DTaP-IPV newborns, Hib, etc.). For the regional level this would be a convex combination of the ten vaccination types and 25 regions, resulting in 250 panels (equivalent for municipal data). More concretely (and ignoring the control variables for brevity sake) we try and estimate \\(y_{ijt}=\\alpha+\\boldsymbol{x}_{it}\\boldsymbol{\\beta}+\\nu_{ij}+\\epsilon_{ijt}\\), where $y_{ijt}$ is the vaccination rate for vaccination $i$ in region $j$ for time $t$. Further, let $\\boldsymbol{x}_{it}$ be the set of Moral Foundations scores for $i$ in $t$. $\\nu_{ij}$ then is the vaccination and region specific error term; it differs between units, but for any particular unit, its value is constant. Within the Netherlands there exists some degree of regional variation as to the general vaccination acceptance, and each type of vaccination also displays some heterogeneity in vaccination rate. By demeaning these fixed effects, we are able to better isolate the true $\\boldsymbol{\\beta}$ without the region- or vaccination-specific fixed effects. We add a clarifying sentence on page 15. (Kindly refer to the ``Response to Reviewers'' pdf for a proper display of the regression symbols.)

Comment 6

Authors should not make strong claims regarding the effect of MF dimensions on vaccine hesitancy on the basis of only 22 sample points and there could be many other factors affecting the hesitancy which have not been considered for the analysis. Also, consider that the moral foundations are not completely independent of each other which is also evident from the MF dictionary words.

Response 6

The first part of this comment seems akin to comment 4 from review round 1 which discusses the issue of overfitting. The subjects in our study have no influence on which version of the brochures they receive, i.e. there is no self-selection by subjects and exposure resembles random assignment. In that sense, we would categorise our research design as a natural experiment in that clusters of subjects have been exposed to different (experimental/control) conditions determined by factors outside the control of us investigators. Among the long tradition of natural experiments, Angrist and Evans [1], for example estimate the effect of family size on the mother's labour market outcomes. Sargent et al. [2] employ a temporary smoking ban in all public spaces in Helena (Montana) to investigate its effect on the rate of heart attacks at the only local hospital. We hope that these examples from a long tradition of literature may convince the reviewer that the 22 different treatment-like realisations in our study actually constitute a comparably fine granulation when comparing with other natural experiments.

This not withstanding, we do acknowledge that vaccine hesitancy is a multi-faceted phenomenon (see, e.g. page 17) and that many different factors play a role in determining the decision whether or not to accept a vaccine. Still, we believe that the power of our study is strong enough to contribute to increasing our understanding of this decision process and some of the aspects it is significantly impacted by. In the Discussion and Conclusion Section we discuss the relationship between the MFT dimensions.

References

1. Angrist JD, Evans WN. Children and Their Parents’ Labor Supply: Evidence from Exogenous Variation in Family Size. The American Economic Review. 1998;88(3):450–477.

2. Sargent RP, Shepard RM, Glantz SA. Reduced incidence of admissions for myocardial infarction associated with public smoking ban: before and after study. BMJ. 2004;328(7446):977–980. doi:10.1136/bmj.38055.715683.55.

---

## [Decision Letter · Decision Letter 2]

20 Oct 2021

Using moral foundations in government communication to reduce vaccine hesitancy

PONE-D-21-07601R2

Dear Dr. Heine,

We’re pleased to inform you that your manuscript has been judged scientifically suitable for publication and will be formally accepted for publication once it meets all outstanding technical requirements.

Kind regards,

Kazutoshi Sasahara

Academic Editor

PLOS ONE

Additional Editor Comments (optional):

Now both reviewers think that all the comments are properly address. Thank you for your careful revision.

Reviewers' comments:

Reviewer's Responses to Questions

**Comments to the Author**

1. If the authors have adequately addressed your comments raised in a previous round of review and you feel that this manuscript is now acceptable for publication, you may indicate that here to bypass the “Comments to the Author” section, enter your conflict of interest statement in the “Confidential to Editor” section, and submit your "Accept" recommendation.

Reviewer #1: All comments have been addressed

Reviewer #2: All comments have been addressed

2. Is the manuscript technically sound, and do the data support the conclusions?

Reviewer #1: Yes

Reviewer #2: Yes

3. Has the statistical analysis been performed appropriately and rigorously? 

Reviewer #1: Yes

Reviewer #2: Yes

4. Have the authors made all data underlying the findings in their manuscript fully available?

Reviewer #1: Yes

Reviewer #2: Yes

5. Is the manuscript presented in an intelligible fashion and written in standard English?

Reviewer #1: Yes

Reviewer #2: Yes

6. Review Comments to the Author

Reviewer #1: (No Response)

Reviewer #2: (No Response)

7. PLOS authors have the option to publish the peer review history of their article (what does this mean?). If published, this will include your full peer review and any attached files.

Reviewer #1: No

Reviewer #2: No

---

## [Editor Report · Acceptance letter]

28 Oct 2021

PONE-D-21-07601R2 

Using moral foundations in government communication to reduce vaccine hesitancy 

Dear Dr. Heine:

I'm pleased to inform you that your manuscript has been deemed suitable for publication in PLOS ONE. Congratulations! Your manuscript is now with our production department. 

Kind regards, 

on behalf of

Dr. Kazutoshi Sasahara 

Academic Editor

PLOS ONE